# Detection through the use of RT-MqPCR of asymptomatic reservoirs of malaria in samples of patients from the indigenous Comarca of Guna Yala, Panama: Essential method to achieve the elimination of malaria

**Lorenzo Cáceres Carrera**[1]*, **Ana María Santamaría**[2], **Anakena Margarita Castillo**[1], **Luis Romero**[3], **Eduardo Urriola**[4], **Rolando Torres-Cosme**[1], **José Eduardo Calzada**[2]

**1** Departmento de Entomología Médica del Instituto Conmemorativo Gorgas de Estudios de la Salud, Ciudad de Panamá, Panamá, **2** Departmento de Parasitología del Instituto Conmemorativo Gorgas de Estudios de la Salud, Ciudad de Panamá, Panamá, **3** Laboratorio Central de Referencia en Salud Pública del Instituto Conmemorativo Gorgas de Estudios de la Salud, Ciudad de Panamá, Panamá, **4** Facultad de Ciencias Biomédicas, Universidad Latina de Panamá, Ciudad de Panamá, Panamá

* lcaceres@gorgas.gob.pa, cacereslorenzo@gmail.com

## Abstract

### Background

*Plasmodium vivax* is the main causative agent of malaria in Panama. However, the prevalence of asymptomatic infections in the different endemic regions remains unknown. Understanding the epidemiological behavior of asymptomatic infections is essential for the elimination of malaria. This study aimed to determine the prevalence of asymptomatic malarial infections in one of the main endemic regions of Panama using multiplex real-time reverse transcription RT-MqPCR.

### Methods

A cross-sectional study was conducted in three communities in the Guna Yala Comarca. A total of 551 thick blood smears and their respective samples on filter paper were collected from volunteers of different ages and sexes from June 20 to 25, 2016. Infections by the *Plasmodium spp.* were diagnosed using microscopy and RT-MqPCR. All statistical analyses were performed using the R software.

### Results

The average prevalence of asymptomatic infections by *P. vivax* in the three communities detected by RT-MqPCR was 9.3%, with Ukupa having the highest prevalence (13.4%), followed by Aidirgandi (11.1%) and Irgandi (3.3%). A total of 74 samples were diagnosed as asymptomatic infections using RT-MqPCR. Light microscopy (LM) detected that 17.6% (13/74) of the asymptomatic samples and 82.4% (61/74) were diagnosed as false negatives. A 100% correlation was observed between samples diagnosed using LM and RT-MqPCR. A

**Data Availability Statement:** All relevant data are within the paper.

**Funding:** This research was funded by the Ministerio de Economía y Finanzas de Panama. Grant No. 9044-052. The funders had no role in study design, data collection and analysis, decision to publish, or preparation of the manuscript.

**Competing interests:** The authors have declared that no competing interests exist.

**Abbreviations: CGY**, Comarca Guna Yala; **DE**, Epidemiology Department; **ICGES**, Instituto Conmemorativo Gorgas de Estudios de la Salud; **LM**, Light microscopy; **MINSA**, Ministerio de Salud; **msnm**, meters above sea level; **NMP**, National Malaria Program; **PCR**, Polymerase chain reaction; **RDTs**, Rapid diagnostic tests; **WHO**, World Health Organization.

total of 52.7% (39/74) of the asymptomatic patients were female and 85.1% (63/74) were registered between the ages of 1 and 21 years. Factors associated with asymptomatic infection were community (aOR = 0.38 (95% CI 0.17–0.83), p < 0.001) and age aOR = 0.98 (95% CI 0.97–1.00), p < 0.05); F = 5.38; p < 0.05).

## Conclusions

This study provides novel evidence of the considerable prevalence of asymptomatic *P. vivax* infections in the endemic region of Kuna Yala, representing a new challenge that requires immediate attention from the National Malaria Program. The results of this study provide essential information for the health authorities responsible for developing new policies. Furthermore, it will allow program administrators to reorient and design effective malaria control strategies that consider asymptomatic infections as a fundamental part of malaria control and move towards fulfilling their commitment to eliminate it.

## Introduction

Malaria is one of the most widespread and serious parasitic diseases in the world [1]. The rapid reduction in malaria worldwide brings us one step closer to eliminating this disease; however, substantial challenges remain. To eliminate and prevent its resurgence, surveillance systems must adapt to the changing epidemiology of malaria and detect all potential malaria infections in a timely manner. Therefore, accurate identification of all types of malaria infections, including symptomatic and asymptomatic, has become a vital component of control and elimination programs [2]. Asymptomatic malarial infection refers to parasitemia of any density in the absence of fever or other acute symptoms in individuals who have not recently received antimalarial treatments [3]. Some asymptomatic infections have levels of parasitemia that are detectable using microscopy, whereas others can only be detected using molecular methods; these are called sub-microscopic infections. At any given time, numerous individuals with detectable malarial parasitemia can be categorized as asymptomatic [4] and are considered important reservoirs for sustaining malaria transmission [5]. However, there are situations in which the presence of asexual forms of the parasite is not associated with clinical malaria; for example, when the level of parasitemia is below the threshold at which an individual develops symptoms (pyrogenic threshold) or when the immune system maintains low parasitemia levels for long periods, the individual is free of symptoms [6]. Furthermore, asymptomatic malaria can occur as a result of the intermittent administration of subtherapeutic levels of antimalarials [7]. Therefore, highly sensitive and low-cost tools for the specific detection of parasites would be useful in the malaria elimination phase [8]. More importantly, for a removal strategy, sensitive and effective detection of infected individuals is necessary and of substantial importance [9, 10].

Strategies aimed at eliminating malaria require increased surveillance and highly sensitive tests that are capable of detecting asymptomatic and low-density infections. These infections are often well below 200 parasites/mL and are important reservoirs capable of transmitting malaria, which must be detected and eliminated [11, 12]. Light microscopy (LM) of thick blood films and Giemsa-stained smears of peripheral blood continue to be considered the gold standard for malaria diagnosis [13]. These microscopic tests, along with rapid diagnostic tests (RDTs) based on the detection of parasitic antigens, are the most commonly used tests in malaria prevalence surveys. Ultrasensitive RDTs improve the detection sensitivity of patients

with parasitemia between 200 parasites/mL and 10,000 parasites/mL [14], but are still limited by their low input volume.

Currently, different molecular tests are used to detect *Plasmodium* species. The sensitivity of polymerase chain reaction (PCR) is due to its ability to detect unique and specific nucleic acid molecules, and the use of concentrated DNA from large samples. The widespread use of PCR-based assays, such as qPCR and multiplex real-time reverse transcription PCR (RT-MqPCR), has revealed new information on malaria prevalence, particularly in low-transmission areas [15, 16]. RT-MqPCR has improved the capacity for the detection of mixed *Plasmodium* infections and *Plasmodium* species in cases of low parasitemia, with a detection limit of less than two parasites/mL, and has the advantage of simultaneous detection of multiple targets in a single run to increase the sensitivity and specificity of the test, compared to LM, RDTs, and nested PCR (nPCR) [17, 18]. In a previous study, in which RT-MqPCR and nPCR were compared to detect *Plasmodium vivax*, RT-MqPCR was more sensitive than nPCR [19, 20].

In Panama, the number of malaria cases has decreased in recent decades, as observed in many countries in the region, reaching an annual average of 647 malaria cases between 2008 and 2018. However, during the last four years there has been a substantial re-emergence of malaria, registering an annual average of 3,818 malaria cases between 2019 and 2022. The total number of malaria cases in 2022 was 7,112 [21], similar to the malaria levels observed 65 years ago at the beginning of the malaria eradication campaign in 1957. To date, there have been no technical reports that have evaluated the main factors responsible for the increase in malaria incidence, nor have there been any recommendations on a strategy that should be followed to tackle the epidemic and reduce the number of cases. To confront this, it is necessary to understand malaria transmission dynamics in endemic sites, mainly within foci with constant transmission, and the potential to cause epidemic outbreaks in different endemic and non-endemic regions of Panama. This depends on the ability to determine, among other important factors, the species of mosquitoes involved in the transmission of malaria, state of resistance to insecticides in vector mosquitoes, infection dynamics at the local level, presence of asymptomatic malaria, and effectiveness of antimalarial strategies. Therefore, studies are needed to generate scientific evidence that can be translated into efficient and effective interventions for malaria control, and thus achieve the elimination of malaria. It is critical to investigate the clinical and public health importance of asymptomatic patients as sources of infection for the maintenance of malaria. Lack of data on asymptomatic malaria parasite reservoir potentially under-estimates burden, undermines efforts for parasite clearance and compromises opportunities for transmission interruption and subsequent efforts to achieve malaria elimination [22]. Thus, we hypothesized that asymptomatic infections represent an important source of infectious gametocytes for vector mosquitoes that maintain malaria transmission in the endemic regions of Panama.

The objective of this study was to determine, through the use of RT-MqPCR, the prevalence of asymptomatic malaria infections in indigenous populations from three communities belonging to Comarca (an administrative region within Panama, assigned as a reserve to a certain indigenous population) of Guna Yala (CGY), which may contribute to the reservoirs of the parasite and maintain malaria transmission. The data obtained in this study will contribute to the improvement and strengthening of policies for malaria elimination by the National Malaria Program (NMP) of the Ministry of Health (MINSA).

## Method

### Selection and description of study sites

The selection of the communities of Irgandi 9˚20'05.66" N; 78˚18'19.49" W; 7 meters above sea level (masl); 300 inhabitants, Ukupa 9˚21'21.80" N; 78˚19'56.21" W; 3 masl; 320 inhabitants

and Aidirgandi 9˚23'05.56" N; 78˚25'12.56" W; 8 masl; 190 inhabitants (Fig 1), was carried out jointly with the NMP of MINSA and based on the information provided on the magnitude of the malaria problem in Panama and particularly in the endemic regions. The three selected communities were located within the CGY in the northeastern part of the country and on the border with Colombia.

Historically, the CGY has been an endemic region for malaria (Fig 2), and, as in the rest of the country, the species of parasitic infection that prevails is *P. vivax*. In the three localities, the residents are 100% indigenous to the Guna ethnic group and had access to malaria diagnosis and treatment through the technical staff of the NMP and health facilities. The CGY has a population of 31,557 inhabitants and comprises approximately 2,340.7 km$^2$ of territorial surface, including approximately 480 km of coastline surrounded by reefs and mangroves and approximately 365 small coral islands, whose ecology has been strongly modified by anthropogenic influences [23]. It has a tropical forest ecology, with lowland areas in the coastal zone that are mainly used by the Guna indigenous population to cultivate coconuts and other crops. The

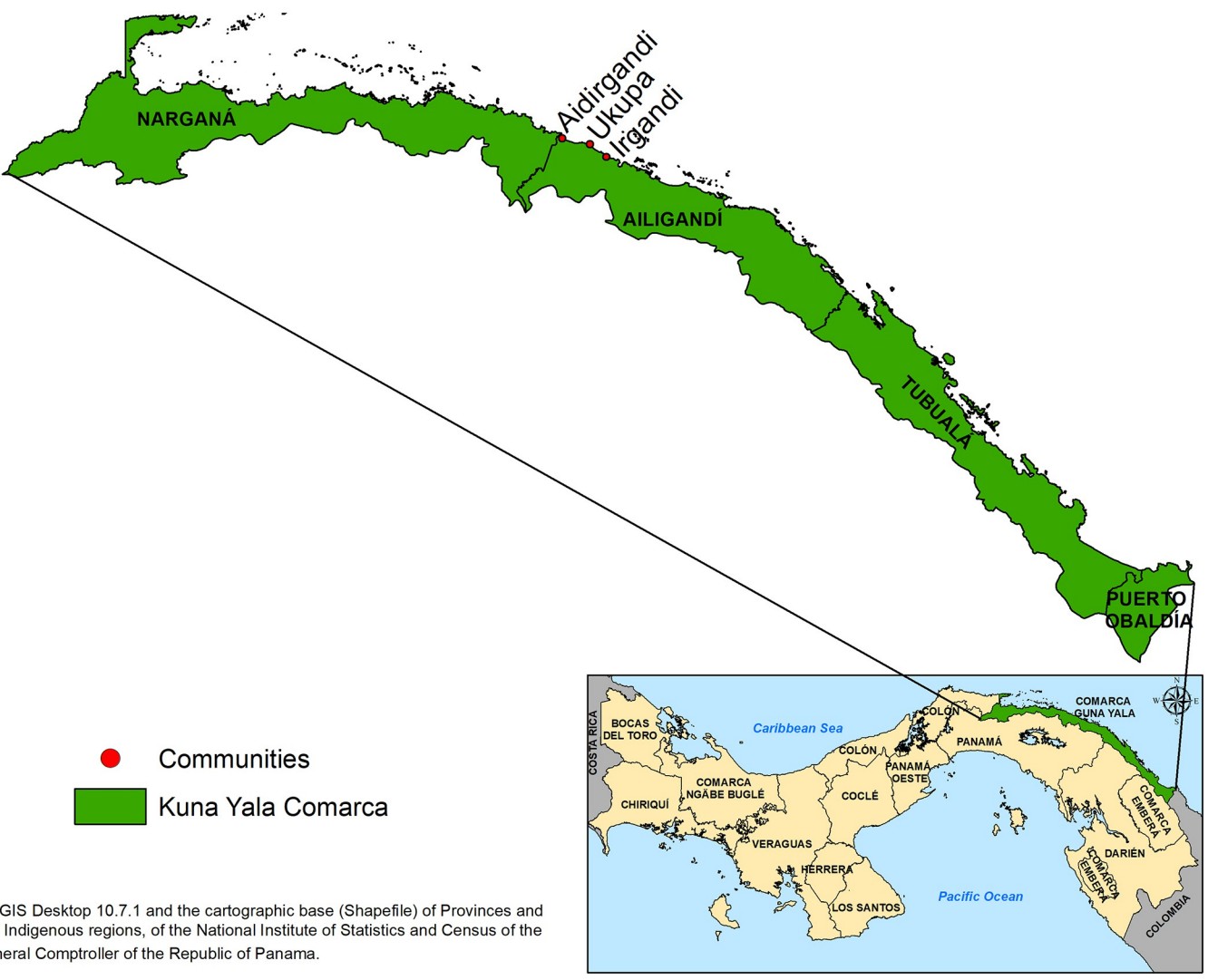

ArcGIS Desktop 10.7.1 and the cartographic base (Shapefile) of Provinces and and Indigenous regions, of the National Institute of Statistics and Census of the General Comptroller of the Republic of Panama.

**Fig 1. Geographic location of the communities studied in the Guna Yala Comarca Panama.**

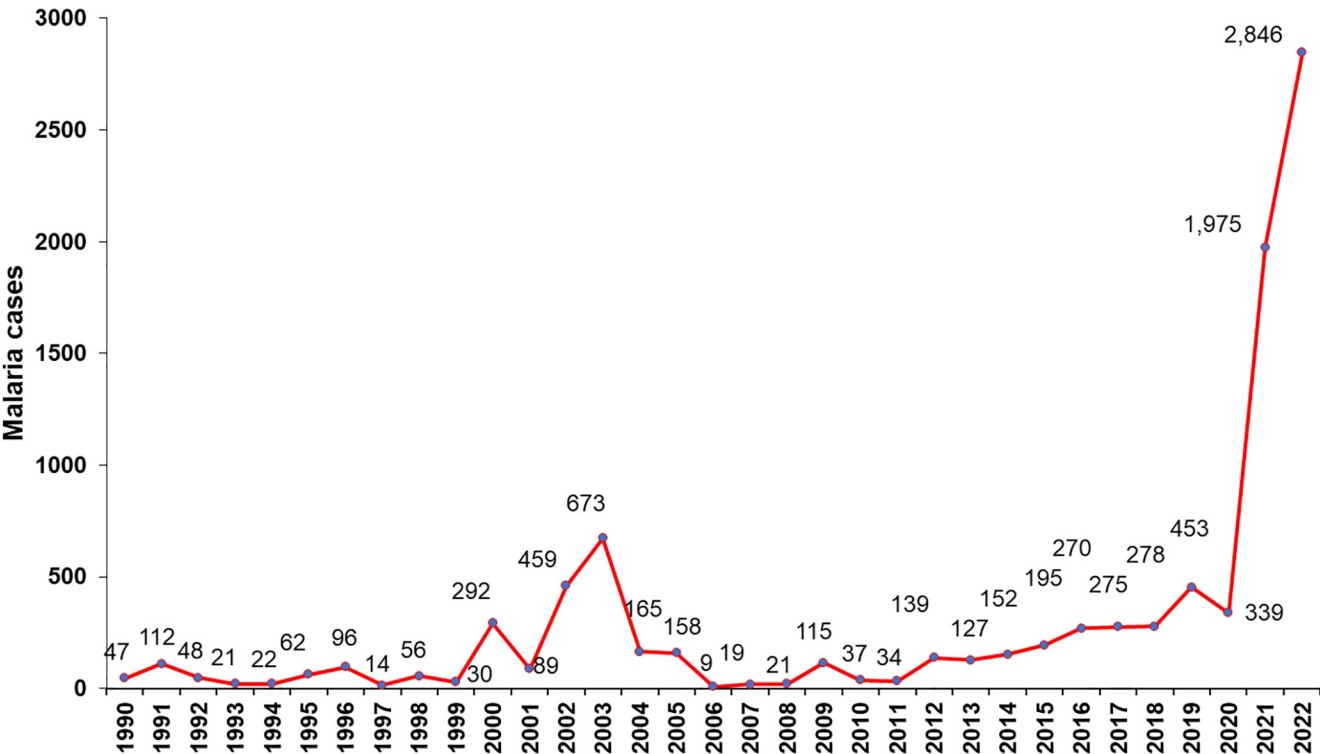

**Fig 2. Epidemiological behavior of malaria cases registered in the Guna Yala Comarca, Panama.** 1990–2022. Source MINSA, Panama.

average monthly temperature in the region is 26 to 27 ˚C, relative humidity between 78 and 90%, and annual rainfall that ranges between 1,600 and 3,000 mm [24]. The CGY region typically exhibits a unimodal rainfall pattern with a drying effect. The dry season lasts from mid-December to April, and the rainy season lasts from May to mid-December.

## Study design

A cross-sectional study was conducted in three selected endemic communities with the collaboration and participation of technical personnel from the NMP of MINSA (Fig 3) to determine the population prevalence of asymptomatic *P. vivax* and *Plasmodium falciparum* malaria parasites. The detection of *Plasmodium ovale* and *Plasmodium malariae* was not considered in this study because they had not been reported in Panama for 41 years [25]. Thick blood samples and filter paper were collected by finger puncture from all inhabitants of different ages and sexes (children and adults) who did not present any clinical signs or symptoms of malaria and had no history of fever in the last 72 h at the time of sampling. Only those who had not left town for at least 15 days were considered. Samples were collected during the period that coincided with the beginning and peak of the malaria season in Panama [26]. All blood samples were obtained from the vector control technical personnel of MINSA.

## Population under study

This study included a sample size that represented a substantial percentage of the population living in each of the three selected communities. A representative sample of 40–50% of the population of each of the selected communities was established, which allowed a greater degree

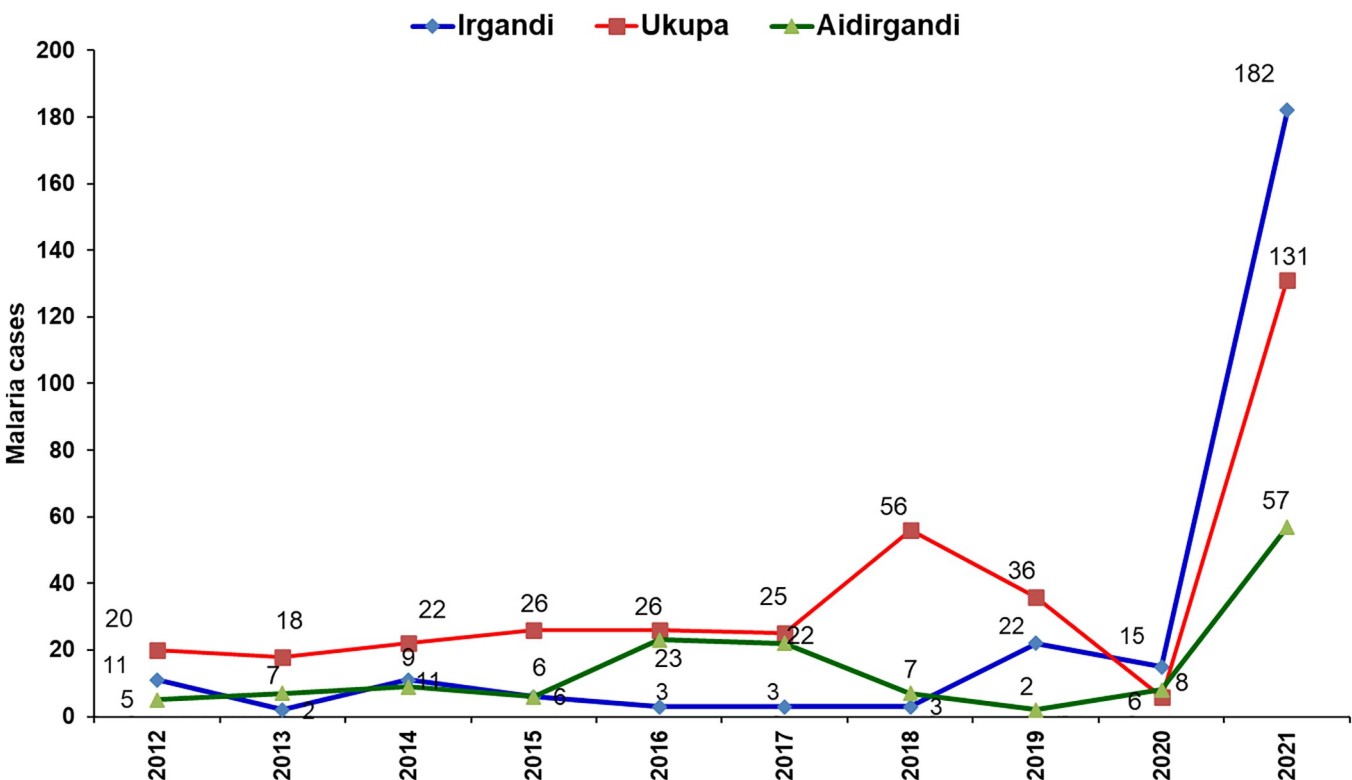

**Fig 3. Malaria cases registered in the communities of Irgandi, Ukupa, and Aidirgandi, Guna Yala Comarca, Panama.** 2012–2021. Source: MINSA, Panama.

of reliability of the results to demonstrate the patterns of asymptomatic people with respect to the rest of the population. The sample included people of different ages and sexes who agreed to participate voluntarily and lived in the selected communities. The children's participation was approved by their parents or tutors. Demographic data, such as age, sex, place of residence, occupation, length of residence in malaria-endemic regions, history of disease, previous episodes of malaria, and antimalarial drugs used in the last month were obtained from the NMP of MINSA. Epidemiological information was obtained from databases provided by the NMP and the Department of Epidemiology of MINSA.

## Operational definition of symptomatic and asymptomatic infections by *Plasmodium spp*

Symptomatic malaria was defined as the presence of malaria-related symptoms (fever $\geq 37.5$ °C, chills, headache, vomiting, and joint pain) within the past two days. For this study, asymptomatic *Plasmodium spp*. infection was defined as the LM detection of asexual stages of *P. vivax* and/or *P. falciparum* in blood samples from individuals who had not received recent antimalarial treatment, which persisted without symptoms during the collection of blood samples. Alternatively, it was defined as the detection of *Plasmodium spp*. DNA by PCR in individuals who remained asymptomatic during blood sample collection. Individuals with negative LM results but positive PCR results were treated with antimalarial agents.

## Exclusion criteria

All asymptomatic individuals were considered to participate voluntarily in the study, with the exception of those who presented the following: 1) patients who had received antimalarial therapy in the last four weeks; 2) individuals with positive LM who remained asymptomatic for 15 days; 3) individuals with symptoms of fever during the last 48 h; 4) participants who reported being sick at the time of sampling; and 5) those who expressed no desire to participate in the study.

## Blood sample collection

This study included both adults and children. The collection of thick blood films, filter paper, and data from the participants was conducted by highly experienced technical personnel from the NMP of MINSA from June 11 to 15, 2016. Blood tests in the three communities were performed on all family members who agreed to participate and did not show signs and symptoms of malaria. Before taking the blood sample, a brief routine questionnaire was carried out to document the travel history and the existence or non-existence of previous episodes of malaria in the last year, while basic information on name, age, and sex was documented by technical personnel from the NMP of MINSA as a routine record when taking a blood sample (average 10 μL) from a person suspected of malaria. Thick blood film and filter paper samples were obtained through digital puncture with a sterile lancet on the participant's index finger. Blood samples were coded, recorded, and carefully stored in sealed bags for subsequent analysis at the Instituto Conmemorativo Gorgas de Estudios de la Salud (ICGES) Department of Parasitology.

## Microscopic diagnosis

All thick blood samples were analyzed by two expert microscopists, each with more than 35 years of experience, who were part of the MINSA NMP. The ability to accurately diagnose blood samples is validated annually by Pan American Health Organization External Performance Evaluation Program for Malaria Microscopic Diagnosis. In addition, personnel conduct the quality control of the diagnostic samples at the national level carried out by the Central Reference Laboratory for Public Health of the ICGES. Malaria is a notable disease in Panama [27]. Each thick-film sample, blood smear, or peripheral smear was stained using Giemsa stain [28]. All samples were observed using an LM, at 1000× magnification. The morphological characteristics of the different stages of *Plasmodium* maturation were the differential criteria for the diagnosis of *P. vivax* and *P. falciparum* using LM. The final diagnosis was made after observing 100 fields before declaring a sample negative or positive for *Plasmodium spp.* [29, 30]. Positive samples after reading 100 fields were converted to parasites/μL of blood [31].

## Diagnosis by RT-MqPCR

*Plasmodium spp.* DNA was extracted from EDTA-treated filter paper blood drops (average 10 μL) using the QIAamp® DNA Blood Mini Kit (QIAGEN, Hilden, Germany), according to the manufacturer's instructions: to a microtube containing 20 μL of protease, 200 μL of blood was added along with 200 μL of AL lysis buffer, with incubation at 56 ˚C for 10 min. The next step was to add 200 μL of ethanol, after which the sample was vortexed and applied to the silica spin column. After centrifugation (6,000 × g for 1 min), the column was inserted into a clean collection tube and the filtrate discarded; 500 μL of wash buffer AW1 was added, and the column was centrifuged (6,000 × g for 1 min). The column was then inserted into a clean

collection tube and the filtrate discarded; 500 μL wash buffer AW2 was added; the column was centrifuged (20,000 × g for 3 min) and the filtrate discarded. A 200 μL volume of AE elution buffer was added to the column, which was then incubated at room temperature for 5 min, centrifuged (6,000 × g for 1 min) and the eluted DNA stored at −20 ˚C. The final volume extracted was 150 μL of each sample.

All DNA samples were analyzed with a commercial RT-MqPCR test with the aim of detecting and differentiating *Plasmodium* species in asymptomatic individuals (FTD Malaria, Fast-Track Diagnostics®, Esch-sur-Alzette, Luxembourg). For the amplifications, 10 μL of the extracted DNA were used and the manufacturer's instructions were followed (FTD Malaria differentiation Manual). The presence of specific sequences was detected by an increase in fluorescence observed from the relevant double-labeled probe and reported as a cycle threshold value (Ct) using an ABI7500 Real-Time Thermal Cycler (Thermo Fisher Scientific).

## Treatment of malaria cases

The supply of antimalarial drugs was rigorously controlled by MINSA. Treatment was provided and strictly supervised by the technical staff of the NMP. All patients who were diagnosed with *P. vivax* infection by microscopy and/or positive PCR received full treatment with 1,500 mg of chloroquine and 210 mg of primaquine for seven days supplied by NMP technical personnel. According to the NMP norm, at 15 days and six months, new blood samples were obtained and examined to confirm that the patient was cured [28].

## Statistical analysis

The prevalence of malaria in each community was presented as a percentage. Considering that there were no previous studies, a sample size of 40% of the population living in each of the three selected communities was considered, a percentage calculated using Epi Info TM version 7.2 program, for the effect of using blood samples. Although a sample size of 40% of the population is sufficient to observe valid patterns with respect to the rest of the population, since the population was relatively small, it was preferred to conduct the survey in ≥50% to have a higher degree of reliability in the results. In the sampling scheme, an attempt was made to have the largest possible number of participants, and thus the entire population had the same probability of being included in the sample; the confidence level was established at 99% and the margin of error at ± 0.03%.

The population was characterized according to basic demographic variables, such as age, sex, place of residence, and time spent in the community. The other variables of interest included a history of malaria infection during the last year, the treatment received, the species of parasite during the last episode, and the results of the microscopic diagnosis. All statistical analyses were performed using R version 4.3.1 (https://www.r-project.org). Possible factors associated with asymptomatic malaria infection were assessed using a multiple logistic regression model with the odds ratio (OR) as a measure of association and the Kruskal–Wallis test. Statistical tests used to determine significance are indicated in Table 3, and p values <0.001 and <0.05 were considered significant.

## Ethical considerations

This study was conducted at the request of the MINSA NMP and approved by the Institutional Technical Committee of the MINSA NMP (Note No. 147/DCV/DGSP/MINSA) and the Guna Yala Health Region (Note No. 25/DCV/DGS/MINSA/2016). The consent and approval of the traditional indigenous authorities (Sahila) of each of the selected communities were obtained through conversations detailing the purpose, benefits, and procedures of the work, which were

explained to each of the Sahilas in the local language (Guna) through a vector control technical staff belonging to this indigenous ethnic group.

## Results

### Analysis of the study population

In total, 551 thick blood smears for microscopic diagnosis and their corresponding samples on filter paper for the molecular diagnosis of *Plasmodium* in asymptomatic individuals were evaluated. The mean age of the participants was 20.3 years and the median was 13 years, with a standard deviation of 17.3. The age range was <1 (4 to 11 months) to 88 years of age, and 65.9% (363/551) of the sampled individuals were between the ages <1–20 years. Within this age group, 37.4% (206/551) corresponded to the <1–10 years group, and 28.5% (157/551) to the 11–20 years group. In total, 56.3% (310/551) of the samples collected from the three selected communities were female.

In the Irgandi community, 179 blood samples were collected (179/551, 32.5%); the mean age was 20.1 years, the median was 13 years, the standard deviation was 18.6, and the age range was <1 (5 to 8 months) to 88 years. A total of 59.2% (106/179) of the samples were female. In Ukupa, a total of 211 blood samples were collected (211/551:38.3%; 95% CI: 34.2–42.4), the mean age was 19.5 years, the median age was 13 years, with a standard deviation of 16.5 and age range of <1 (4 to 11 months) to 69 years. In total, 52.1% (110/211) of the samples were from males. In Aidirgandi, a total of 161 blood samples were collected (161/551:29.2%; 95% CI: 25.4–33.0), the mean age was 20.6 years, the median 13 years, with a standard deviation 17 and age range of <1 (4 to 7 months) to 75 years. A total of 59% (95/161) of participants were female. Tables 1–3 shows the distribution of blood samples obtained from the population by locality. In Irgandi 60% of blood samples were obtained (179/300 inhabitants; 95% CI: 52.8–67.2), in Ukupa it was 66% (211/320 inhabitants; 95% CI: 59.6–72.4); and in Aidirgandi it was 85% (161/190 inhabitants; 95% CI: 79.4–90.5).

### Microscopic diagnosis

All smears (n = 551) from asymptomatic individuals were analyzed using LM. Thirteen cases were asymptomatic for *P. vivax*, accounting for 2.4% of the cases detected (13/551). LM diagnosed 17.6% (13/74) of asymptomatic cases detected using RT-MqPCR. No asymptomatic *P. falciparum* infection was detected in blood samples. The parasitemia of the asymptomatic cases

**Table 1. Number of blood samples taken by age group and sex for the diagnosis of asymptomatic malaria by LM and RT-MqPCR in the community of Irgandi, Guna Yala Comarca, Panama.**

| Age Group | Irgandi (N = 300 habitants) | | | | |
|---|---|---|---|---|---|
| | **Male** | **Female** | **Total Samples** | **% Samples** | **% Samples comm** |
| < 1 a 10 | 35 | 35 | 70 | 39.1 | 60% |
| 11 a 20 | 16 | 29 | 45 | 25.1 | |
| 21 a 30 | 9 | 14 | 23 | 12.8 | |
| 31 a 40 | 4 | 8 | 12 | 6.7 | |
| 41 a 50 | 2 | 6 | 8 | 4.5 | |
| 51 a 60 | 3 | 7 | 10 | 5.6 | |
| 61 a 70 | 3 | 5 | 8 | 4.5 | |
| > 71 | 1 | 2 | 3 | 1.7 | |
| **Total** | **40.8% 73/179** | **59.2% 106/179** | **32.5% 179/551** | **100** | **179/300** |

Childrens <1 year (4 to 11 months); cmm: community; LM: light microscopy

**Table 2. Number of blood samples taken by age group and sex for the diagnosis of asymptomatic malaria by LM and RT-MqPCR in the community of Ukupa, Guna Yala Comarca, Panama.**

| Age Group | Ukupa (N = 320 habitants) | | | | |
|---|---|---|---|---|---|
| | Male | Female | Total Samples | % Samples | % Samples comm |
| < 1 a 10 | 42 | 34 | 76 | 36 | 66% |
| 11 a 20 | 42 | 29 | 71 | 33.6 | |
| 21 a 30 | 9 | 12 | 21 | 10 | |
| 31 a 40 | 3 | 9 | 12 | 5.7 | |
| 41 a 50 | 7 | 10 | 17 | 8.1 | |
| 51 a 60 | 3 | 5 | 8 | 3.8 | |
| 61 a 70 | 5 | 1 | 6 | 2.8 | |
| > 71 | 0 | 0 | 0 | 0 | |
| **Total** | **52.6% 111/211** | **47.4% 100/211** | **38.3% 211/551** | **100** | **211/320** |

Childrens <1 year (4 to 11 months); cmm: community; LM: light microscopy

**Table 3. Number of blood samples taken by age group and sex for the diagnosis of asymptomatic malaria by LM and RT-MqPCR in the community of Aidirgandi, Guna Yala Comarca, Panama.**

| Age Group | Aidirgandi (N = 190 habitants) | | | | |
|---|---|---|---|---|---|
| | Male | Female | Total Samples | % Samples | % Samples comm |
| < 1 a 10 | 35 | 25 | 60 | 37.3 | 85% |
| 11 a 20 | 19 | 22 | 41 | 25.5 | |
| 21 a 30 | 1 | 17 | 18 | 11.2 | |
| 31 a 40 | 7 | 13 | 20 | 12.4 | |
| 41 a 50 | 2 | 6 | 8 | 5 | |
| 51 a 60 | 1 | 10 | 11 | 6.8 | |
| 61 a 70 | 0 | 0 | 0 | 0 | |
| > 71 | 1 | 2 | 3 | 1.8 | |
| **Total** | **41% 66/161** | **59% 95/161** | **29.2% 161/551** | **100** | **161/190** |

Childrens <1 year (4 to 11 months); cmm: community; LM: light microscopy

diagnosed by microscopy significantly varied according to the parasite density, being classified between low (26 to 113 parasites/µL) to moderate (1,541 to 3,571 parasites/µL) according to the level of parasitaemias, Overall, the median parasite load in the study population was 113 parasites/µL with a range of 26 to 3,571 parasites/µL blood. Most of the asymptomatic individuals showed very low parasitemia and consequently low parasite counts (<500 parasites/µL of blood). Among those diagnosed as positive using microscopy, five blood samples presented gametocytes, with a median of 20 gametocytes (range, 3–30 gametocytes), and 61.5% presented mild parasitemia (Table 4). Using RT-MqPCR as the gold standard, LM detected only one-sixth of asymptomatic infections. The highest percentage of asymptomatic infections detected by LM was in patients between the ages of 1 and 12 years (n = 11, 84.6%); the mean age was 11.8, the median was 10 years (range, 1–37 years), and 61.5% (n = 8) males were e diagnosed using LM.

## Diagnosis by RT-MqPCR

All samples (n = 551) were analyzed using RT-MqPCR. In the three communities studied, the percentage of *P. vivax* infection detected in the inhabitants using RT-MqPCR was 13.4% (74/

**Table 4. Parasite densities (parasites/µL) observed in asymptomatic *P. vivax* cases diagnosed by optical microscopy in the Guna Yala Comarca, Panama.**

| Variable | Frequency (Median/Range Parasites/µ) | Samples with Gametocytes Median (Range) | Porcentaje (%) |
|---|---|---|---|
| Asymptomatic infections by density | | | |
| Mild parasitaemia (< 1000 parasites/µl) | 8 (50.5 / 26 to 113) | | 61.5 |
| Moderate parasitaemia (1000–9999 parasites/µl) | 5 (2,631 / 1,541 to 3,571) | 5 (20, 3 to 30) | 38.5 |
| Severe parasitaemia (≥ 10000 parasites/µl) | | | |
| Positive (95% CI) | | 13 | 17.6 (8.9–26.3) |

551), which was much higher than that detected using LM (2.4%, 13/551). The highest prevalence of asymptomatic infections by *P. vivax* detected using RT-MqPCR was in Ukupa at 13.4% (43/320 inhabitants), while 2.2% (7/320 inhabitants) was detected using LM. Aidirgandi followed at 11.1% (21/190 inhabitants) using RT-MqPCR, and 3.2% (6/190 inhabitants) using LM. Finally, in Irgandi, the lowest prevalence of 3.3% (10/300 inhabitants) was detected using RT-MqPCR. No asymptomatic cases were detected using LM in this community (Table 5). Asymptomatic *P. falciparum* infections were not detected using RT-MqPCR or LM.

The Ukupa community had the largest number of asymptomatic individuals with *P. vivax* detected using RT-MqPCR, with 43 cases (43/74, 58.1%), and 51.2% (22/43) of the positive samples being female. In Aidirgandi, 52.4% (11/21) of samples were asymptomatic for *P. vivax*. Finally, in the Irgandi community, females represented 60% (6/10) of the positive samples from asymptomatic individuals (Table 5). In Irgandi, no cases of asymptomatic *P. vivax* infection were diagnosed using LM.

It was not possible to identify individuals with malarial symptoms at the time of sampling. None of the participants had left their communities during the last 20 days. RT-MqPCR showed an increase in the proportion of positivity compared with that using LM, registering almost six times more positivity than LM (1:5.7). In the distribution by age, the mean and median age was very similar in the communities of Irgandi (12.3 years and 9 years) and Ukupa (13.8 years and 9 years), while those of Aidirgandi showed differences with a mean of 29.3 years and median of 13 years, respectively. A total of 79.7% of cases diagnosed by RT-MqPCR were between the ages of 1 and 40 years and 74.3% were under 14 years of age (1 to 14 years). Through the use of the multiple logistic regression model, it was determined that the factors associated with asymptomatic infection were the community with aOR = 0.38 (95% CI 0.17–

**Table 5. Diagnosis by LM and RT-MqPCR of asymptomatic people with *P. vivax* in the Guna Yala Comarca in Panama during 2016.**

| | N | Irgandi (N = 179) | Ukupa (N = 211) | Aidirgandi (N = 161) | Test statistic |
|---|---|---|---|---|---|
| Asymptomatic cases by community (%) | 74 | 10/179 (5.6%) | 43/211 (20.4%) | 21/161 (13.0%) | aOR = 0.38 (95% CI 0.17–0.83), p < 0.001[1] |
| Female gender (%) Age ($\bar{x}$) | 74 | 6/10 (60%) | 22/43 (51.2%) | 11/21 (52.4%) | aOR = 0.98 (95% CI 0.97–1.00), p < 0.05[1] |
| | | | | | F = 5.38, p <0.05[2] |
| Median age in years (IQR) Range | | 9 (4 to 29) | 9 (1 to 67) | 13 (2 to 75) | |
| Microscopy diagnostic (% positive, 95% CI) | 13 | | 9.5% (41.5–106.5) | 8.1% (38.9–109.1) | |
| Median parasites/µl (IQR) Range | | 0 | 113 (38 to 3,571) | 173 (26 to 3,571) | |
| RT-MqPCR (% positive, 95% CI) | 74 | 13.5% (5.7–21.3) | 58.1% (66.8–69.3) | 28.4% (18.1–38.7) | |
| Prevalence (%), (95% CI) | | 3.3% (1.3–5.3) | 13.4% (9.7–17.1) | 11.1% (6.6–15.6) | |

LM: light microscopy,

[1]Logistic regression (aOR),

[2]Kruskal–Wallis

0.83), p<0.001, and age aOR = 0.98 (95% CI 0.97–1.00), p<0.05 and F = 5.38, p<0.05 (Table 3). Sex was not significantly associated with asymptomatic infection.

## Discussion

### Incidence of malaria in CGY

In the CGY, one of the most important variables that determines the intensity and prevalence of malaria is the internal migration of native people within communities in the region and the arrival of immigrants from South America who are symptomatic and asymptomatic carriers of the parasite to communities that are receptive and vulnerable to malaria transmission and outbreaks. Malaria transmission dynamics are characterized by seasonal origins through small foci or epidemic outbreaks of transmission, whose incidence is closely related to migratory movements, unprotected homes, anopheline breeding sites near homes, climate, and environmental, socioeconomic, and cultural factors. The population has a low collective immunity; therefore, there is a high probability of epidemics [26]. The CKY region has historically been endemic for malaria, where *P. vivax* infections and periodic appearances of *P. falciparum* predominate, mainly in imported cases in recent years. From 1965 to 2022, this region has contributed 15.1% of the total cases registered throughout the country [21]. The results of this study demonstrate the existence of asymptomatic infections by *P. vivax*, for the first time in one of the main endemic regions in the country. *P. vivax* was the only infectious species detected in the asymptomatic cases in this study. This species is the main parasite that causes malaria in Brazil [32]. From 2000 to 2022, 42,683 cases of malaria were registered throughout the country, and 92.1% of the reported cases were caused by *P. vivax* [21]. *Plasmodium vivax* is the predominant parasite species causing malaria in the Americas. It is genetically and geographically extremely diverse, with substantial genetic divergence between populations separated by relatively short distances [33, 34].

The high prevalence of *P. vivax* asymptomatic individuals observed in the CGY may explain the maintenance or persistence of residual foci of transmission owing to the existence of asymptomatic human reservoirs with parasitemia that are not regularly detected using LM. This was unrecognized before because malaria diagnosis was only conducted using LM. As shown in this study, LM detected only a small fraction of malaria-positive cases. These findings dispute the historical hypothesis that symptomatic and asymptomatic immigrants from South America are responsible for the constant prevalence of malaria in this region. This study demonstrates that asymptomatic infected native populations may be involved in the local maintenance of malaria transmission. People with undetected asymptomatic infections do not seek care from health services or NMP staff and do not receive antimalarial treatment, thus contributing to the maintenance of the transmission cycle through the participation of anopheline vectors [26].

Asymptomatic individuals with submicroscopic parasitemia may sustain continuous malaria transmission, even under low transmission conditions, if they are not diagnosed using a highly sensitive method and promptly treated with effective antimalarial drugs [35, 36]. The detection of asymptomatic infections [37] and the demonstration that asymptomatic people carrying parasites can infect vector anopheline mosquitoes [38], raises concerns that those asymptomatic people may be playing a bigger role in the dynamics of transmission and maintenance of the disease. Consequently, the treatment of asymptomatic and submicroscopic infections can be of fundamental importance for improving antimalarial strategies and moving towards their elimination [36]. Asymptomatic parasitemia is considered relatively uncommon in areas with low seasonal transmission that predominantly occur in Asia and America. Recent epidemiological studies have frequently used sensitive molecular methods to estimate the true

burden of malaria in endemic areas to plan effective control methods for this disease [39–41]. Thus, the availability of an efficient and effective technique to detect asymptomatic infections may be a key factor in decision-making by NMP in the fight against elimination of malaria. There is a need for a strategy to detect and reduce the number of asymptomatic cases and submicroscopic infections to eliminate malaria in endemic regions. This observation agrees with reports from similar studies conducted in Sri Lanka [42, 43], Iran [44], the Solomon Islands [45], and Paraguay [46], which suggest the importance of reducing asymptomatic cases, submicroscopic parasitemia, and symptomatic cases of malaria in at-risk populations to achieve the elimination goal.

## Analysis of the studied population

A high percentage of blood samples taken from the three communities corresponded to those aged <1–20 years (n = 363, 65.9%). Within this age group, a substantial percentage of the blood samples were collected from the ages of <1–10 years (n = 206, 37.4%). This group corresponded with the significant prevalence of asymptomatic individuals with *P. vivax* infection diagnosed using RT-MqPCR in the three communities (38/810, 4.7%). Regardless of the age range with the highest prevalence, this study revealed the silent presence of asymptomatic reservoirs of the *P. vivax* malarial parasite in different age groups in the three communities. Among demographic factors, female gender and children in the <1–10 age group were associated with asymptomatic *P. vivax* infections, representing 51.4% of all diagnosed asymptomatic cases. This indicates that the greatest risk of contracting asymptomatic malaria was found within this age group, and the probability decreased with increasing age. It was observed that 27.3% of children positive for malaria were under 10 years of age, representing a significant percentage of the incidence of the disease [26]. These results are consistent with observations in Ethiopia and Peru [47, 48]. In Africa, school-aged children are more likely to develop *P. vivax* malaria [49, 50]. This population has a high probability of being exposed to competent vectors and high risk of acquiring malaria. This asymptomatic age group poses many problems because they are unknown carriers of the parasite and these reservoirs contribute to the maintenance of the life cycle of the parasite and disease transmission, as documented in malaria-endemic communities [51, 52]. Notably, all age groups were exposed to asymptomatic *P. vivax* infections, and other similar studies confirm this pattern. The higher prevalence of asymptomatic infections in underage children could be the result of increased exposure to *Plasmodium* due to the inappropriate use of bed nets and antimalarial treatment [53, 54].

The population studied was relatively stable and resided for many years in the three communities studied. The three communities are located on the seashore and are close to rivers or streams, with diverse anopheline vector breeding sites. The distribution of asymptomatic individuals showed similar spatial behavior in the three communities, with a slight concentration in the central part of the communities studied. The remaining asymptomatic individuals were dispersed throughout each community. This may explain the similar historical epidemiological behavior of malaria in the three communities, which may be due to their proximity. The entomological variables that could explain this distribution pattern are yet to be studied. Epidemiologically, malaria has been associated with factors related to the vector, parasite, human host, and environment. However, only human host-related factors were investigated in this study. Several studies have reported human behavior and practices that are known to increase the risk of malaria transmission [55, 56]. In this study, the two factors associated with asymptomatic malaria infection were community and age. This may be due to the permanence of adult women and children in homes, the activities carried out in the intra- and peri-domestic environments during the period of highest frequency of bites of anopheline vectors, and the

proximity of the breeding sites located in rivers and streams near these communities. The high rate of asymptomatic malaria infections in women and children may be due to exposure to mosquito bites in various environments, densities of adult mosquitoes, location of houses in relation to the breeding sites of mosquitoes, low immunity, unprotected housing, lack of knowledge, and negative health behaviors [57–59]. Based on the results obtained, the probability of being asymptomatic due to malaria was higher in children than that in adults (n = 19, 25.7%), suggesting that age could be an important risk factor for the presence of asymptomatic infections. Similarly, other studies have reported that children can act as parasite reservoirs and facilitate malarial transmission. This can have serious implications in underaged children as it can affect brain development, academic performance [60, 61], and parasitemia, leading to severe malaria [62, 63]. Therefore, it is necessary to predominantly direct malaria control intervention initiatives to underage children to achieve malaria control throughout the entire CGY.

## Microscopic diagnosis

Accurate diagnosis of *Plasmodium* species is important to establish the correct antimalarial treatment as well as apply effective malaria control strategies in endemic regions where there is more than one species of malaria parasite, such as the CGY. Considering the indications of the WHO, malaria control requires a high-quality diagnostic method to detect parasite species before prescribing antimalarial treatments. The parasitological diagnosis of malaria indicates the selection of adequate treatment, supports the characterization of the response to treatment, and allows for early identification of the parasite and timely treatment [64].

Microscopic diagnosis detected only 17.6% of the asymptomatic *P. vivax* malaria cases diagnosed using RT-MqPCR in the three communities. LM misses approximately 50% of all PCR-positive malaria infections in a previous study [65]. The microscopic diagnosis varies considerably between <10–500 parasites/µL of blood. Microscopy is only reliable if there is a high density of parasites. In the presence of low parasite densities, a negative result does not exclude the presence of malaria infection [9, 66]. The effectiveness of LM for malaria diagnosis depends on the maintenance of a high level of staff competence and accurate test performance [67].

Among the individuals with positive asymptomatic infection diagnosed using microscopy, five had gametocytes in low numbers (3–30 gametocytes). These sexual stages are important for the maintenance of malarial transmission. In studies conducted on individuals with *P. vivax* gametocytes, the proportion of asymptomatic individuals was low. This may be explained by the low probability of gametocyte detection at low blood-stage parasite densities, as observed in asymptomatic individuals. Although cases with lower parasitemia and, therefore, lower gametocytemia have been detected, asymptomatic individuals are capable of infecting mosquitoes [37] and, therefore, could be an important reservoir, especially for seasonal outbreaks [68]. A study in the south of the Amazon estimated that 54.4% of all parasite biomass belonged to asymptomatic individuals, accounting for 56.6% of all infections [69]. Other reports have shown a higher rate of gametocyte carriers; however, these were symptomatic patients with high parasite densities, which correlated with higher gametocyte detection [70–72]. Malaria is a public health problem in the CGY; *P. vivax* parasitic infections account for more than 90% of malaria cases [26, 27]. Previous studies have shown that parasite density decreases with age [73]. The density of the parasite in this study was between low and moderate (26 to 3,571 parasites/µL).

This study highlights the low effectiveness and possible deficiencies in the diagnosis of asymptomatic malaria using LM. If PCR is considered the gold standard for comparing the results obtained using LM, a high percentage of false negatives (82.4%) was observed in this

study. False-negative LM results are known to increase as parasite density decreases [74]. False negatives are a major public health problem because some individuals return home without the correct diagnosis and treatment. The low density of parasites, together with the high number of false negatives detected using microscopy, indicate that it is often difficult to obtain a good microscopic diagnosis [75]. Although LM has been recommended by the WHO as the gold standard for the diagnosis of symptomatic patients, many studies have reported that at low densities, especially among asymptomatic carriers, parasites are easily missed by this method because of inherent limitations, such as lack of experience in microscopic diagnosis, sample preparation methods, staining techniques, and detection limits [76–79]. Despite these limitations, microscopic diagnosis using thick blood film and peripheral blood smears remains the "gold standard" for the diagnosis of parasites in NMP [27, 80].

## Diagnosis by RT-MqPCR

Notably, PCR is not applied as a routine diagnostic method by public health services in Panama. The use of molecular techniques by public health services can substantially contribute to the detection of submicroscopic and asymptomatic infections that cannot be detected using LM, which is used as a standard method by the NMP. In a previous study conducted on the characterization of a recent malaria outbreak in the CGY, it was suggested that symptomatic and asymptomatic immigrants from South America and natives of the Guna population could participate in this epidemic and in the maintenance of transmission. The use of sensitive molecular methods was recommended to detect people with asymptomatic infections, and thus determine the actual burden of malaria [26]. To address this problem, health policy-makers must develop collaborative strategies with scientific centers and institutions to detect asymptomatic cases of malaria to achieve effective control and elimination. The results of this study highlight the need for NMP to incorporate more sensitive malaria diagnostic methods, such as the use of molecular tests, to characterize endemic and non-endemic regions.

In the Americas, reports of asymptomatic infections have been relatively recent. In Brazil, 70% of mineral prospectors in Mato Grosso are asymptomatic [81]. Among indigenous populations in the Colombian part of the Amazon basin, 21.6% were asymptomatic [82, 83]. In the Peruvian part of the Amazon basin, a prevalence of 17.6% was reported [84], and in the middle and upper reaches of the Río Negro, 20.4% of the cases among the population involved in mining activities were asymptomatic [85]. In contrast, a study carried out in Venezuela detected a prevalence of infection of 1.3% using RDTs and 8% using LM, and PCR detected 10 asymptomatic cases, with a sensitivity and specificity of 100% and 93.4%, respectively [86]. In Colombia, a prevalence of asymptomatic patients detected using qPCR of 10% in 2011, 7% in 2013, and 4% in 2014, respectively, was observed, while by LM the prevalence was <0.2%. *P. vivax* was predominant in the present study [87]. In Peru, regarding cases detected using qPCR and RT-MqPCR, 59.8% of all *P. vivax* gametocyte carriers were asymptomatic and 31.9% were submicroscopic, respectively [88].

To achieve effective control and move towards the elimination of malaria in Panama, precise epidemiological characterization of symptomatic and asymptomatic malaria carriers and their geographical distribution is necessary. These data are essential for developing effective strategies against disease transmission in endemic and non-endemic regions. Once the results were obtained through diagnosis using LM and RT-MqPCR for individuals with asymptomatic *P. vivax* infection, NMP was notified for treatment and follow-up. All asymptomatic patients were successfully treated, and each case was followed up according to the national standard for the therapeutic management of patients with *P. vivax* malaria [27]. The same epidemiological scenario observed in the CGY may occur in different endemic regions of the

country with active malaria transmission. Therefore, this situation brings a new paradigm for the NMP that requires immediate attention to maintain the achievements obtained in the control of malaria to date and establish the new goal of eliminating the disease in the country. To the best of our knowledge, to date, asymptomatic infections had never been scientifically described in Panama; they were only mentioned as a possible immunological state that people living in different endemic regions developed [26, 89]. These results support the need for more sensitive diagnostic techniques to detect asymptomatic malaria.

There is a need to conduct new studies on CGY that include entomological aspects and evaluate the immune response for a better understanding of asymptomatic and symptomatic infections by *Plasmodium spp*. and their contribution to the dynamics of malaria transmission. Similarly, it is necessary to conduct studies in other endemic regions of the country to establish the potential role of asymptomatic infections in the general prevalence of malaria, genotyping of asexual and sexual parasites, seasonality, variability, and their contribution to the incidence of malaria in each endemic region of the country. All evidence generated must be delivered to the NMP of MINSA so that they consider the threat that asymptomatic malaria represents, which can be translated into effective interventions for control and elimination of malaria in the country.

## Limitations

The current study has several limitations. First, it had limited geographical coverage; therefore, generalization of the prevalence of asymptomatic infections in different endemic regions was not possible. Furthermore, the study was based on a cross-sectional design, and therefore had limitations in establishing any type of causal association between asymptomatic infections and other factors. No short-term follow-up was conducted to assess whether asymptomatic individuals progressed to symptomatic malaria as other studies have done [90, 91]. However, despite these limitations, the study presents important information on the disease burden and possible risk factors that may be involved with asymptomatic infections, which could be very useful for the formulation of effective strategies for surveillance, prevention and control of malaria transmission and its elimination.

## Conclusion

This study provides novel evidence of the considerable prevalence of asymptomatic *P. vivax* infections in the endemic region of Guna Yala, representing a new challenge that requires immediate attention from the NMP. The results of this study provide essential information for the health authorities responsible for developing new policies. Furthermore, it will allow program administrators to reorient and design effective malaria control strategies that consider asymptomatic infections as a fundamental part of malaria control and move towards fulfilling their commitment to eliminate it. This study highlights the need for longitudinal studies conducted in all endemic regions of Brazil to evaluate the prevalence of asymptomatic infections.

## Acknowledgments

We would like to thank Jose Lasso (RIP), Chief of the Department of Vector Control from the Ministry of Health; Carlos Victoria (RIP), Director of the National Malaria Program; Luis De Urriola Director of CGY Health Region; Maibel Pérez Medical Director of the Playón Chico Health Center; Cipriano Ayerza regional coordinators of vector control; vector control technicians; Manuel Herrera, Eliverio López, Humberto Pérez, Arniel López, Zacarias Nelson, and Lester Porras. To the traditional indigenous authorities: Valerio Grimaldo the Ukupa, Lizandro López the Irgandi and Rolando Jimenez the Aidirgandi (RIP). BS. Dianik Moreno, head of

the Parasitology and Malaria Section of the Central Reference Laboratory in Public Health for the collaboration provided; Dr. José Luis Ramirez for reviewing the manuscript; and BS. Alberto Cumbrera, for the development of maps, and all operative personnel, collaborated and supported the development of this research.

## Author Contributions

**Conceptualization:** Lorenzo Cáceres Carrera.

**Data curation:** Lorenzo Cáceres Carrera, José Eduardo Calzada.

**Formal analysis:** Lorenzo Cáceres Carrera, Ana María Santamaría, Anakena Margarita Castillo, Luis Romero, Eduardo Urriola, Rolando Torres-Cosme, José Eduardo Calzada.

**Funding acquisition:** Lorenzo Cáceres Carrera.

**Investigation:** Lorenzo Cáceres Carrera, Ana María Santamaría, Anakena Margarita Castillo, Eduardo Urriola, Rolando Torres-Cosme, José Eduardo Calzada.

**Methodology:** Lorenzo Cáceres Carrera, Ana María Santamaría, Anakena Margarita Castillo, Luis Romero, Eduardo Urriola, Rolando Torres-Cosme, José Eduardo Calzada.

**Project administration:** Lorenzo Cáceres Carrera.

**Supervision:** Lorenzo Cáceres Carrera.

**Visualization:** Lorenzo Cáceres Carrera.

**Writing – original draft:** Lorenzo Cáceres Carrera.

**Writing – review & editing:** Lorenzo Cáceres Carrera, Ana María Santamaría, Anakena Margarita Castillo, Luis Romero, Eduardo Urriola, Rolando Torres-Cosme, José Eduardo Calzada.

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
