## [Decision Letter · Decision Letter 0]

4 Oct 2023

PONE-D-23-20941Detection through the use of RT-MqPCR of asymptomatic reservoirs of malaria in samples of patients from the indigenous Comarca of Guna Yala, Panama: Essential method to achieve the elimination of malaria.PLOS ONE

Dear Dr. Cáceres Carrera, Thank you for submitting your manuscript to PLoS ONE. After careful consideration, we felt that your manuscript requires substantial revision, following which it can possibly be reconsidered, thus governing the decision of a “major revision”. As requested by the reviewers, the structure and content of the manuscript need to be deeply revised. The authors need to revise the text making it more concise and (considerably) shorter, eliminating unnecessary repetitions and removing far-fetched conclusions. In addition, the authors need to address several concerns, particularly related to the data available (raw data), statistical analysis, methods and results. Finally, we strongly suggest a professional copy-editing service. For your guidance, a copy of the reviewers' comments was included below.

We look forward to receiving your revised manuscript.

Kind regards,

Luzia H Carvalho, Ph.D.

Academic Editor

PLOS ONE

A clean copy of the edited manuscript (uploaded as the new *manuscript* file).

6. We note that Figure 1 in your submission contain [map/satellite] images which may be copyrighted. All PLOS content is published under the Creative Commons Attribution License (CC BY 4.0), which means that the manuscript, images, and Supporting Information files will be freely available online, and any third party is permitted to access, download, copy, distribute, and use these materials in any way, even commercially, with proper attribution. For these reasons, we cannot publish previously copyrighted maps or satellite images created using proprietary data, such as Google software (Google Maps, Street View, and Earth). For more information, see our copyright guidelines: http://journals.plos.org/plosone/s/licenses-and-copyright.

7. We note that Figure 2 in your submission contain copyrighted images. All PLOS content is published under the Creative Commons Attribution License (CC BY 4.0), which means that the manuscript, images, and Supporting Information files will be freely available online, and any third party is permitted to access, download, copy, distribute, and use these materials in any way, even commercially, with proper attribution. For more information, see our copyright guidelines: http://journals.plos.org/plosone/s/licenses-and-copyright.

1. You may seek permission from the original copyright holder of Figure 2 to publish the content specifically under the CC BY 4.0 license.

Reviewers' comments:

Reviewer's Responses to Questions

**Comments to the Author**

1. Is the manuscript technically sound, and do the data support the conclusions?

Reviewer #1: Yes

Reviewer #2: Yes

2. Has the statistical analysis been performed appropriately and rigorously? 

Reviewer #1: Yes

Reviewer #2: No

3. Have the authors made all data underlying the findings in their manuscript fully available?

Reviewer #1: No

Reviewer #2: Yes

4. Is the manuscript presented in an intelligible fashion and written in standard English?

Reviewer #1: No

Reviewer #2: No

5. Review Comments to the Author

Reviewer #1: The work presents a topic of extreme relevance for understanding the epidemiology of malaria.

However, it is necessary to make major changes to the text to better understand the work.

I would like to see the raw population data and the RT-MqPCR results, with the sample amplification curves. To better analyze the results.

I suggest passing the article through an English reviewer.

59: Confirm the number of positives in the study. In the text this value is 13.4%. reconfirm this value.

60: Confirm the number of positives in Irgandi. In the text this value is 1.8%. reconfirm this value.

65: To better characterize the age of your population, it would be extremely important to write the age in months of children under one year old. I suggest changing the minus sign to x months until 21 years.

101: cite a reference for this sentence.

111: Wouldn't that be a strategy for eradicating and eliminating malaria?

117: Change the parasites/ml for parasites/ul. parasites/ul is the most used unit.

128: add Polymerase chain reaction (PCR).

135: change for parasites/μl.

137: I couldn't understand what an nPCR is and what its target is.

163: It would be interesting to reference studies that demonstrate the role of asymptomatic people in the transmission of malaria.

194: Throughout the text the degree sign Celsius is written with a subscript. Change to °.

203: The sentence is incomplete, please add "malaria parasites".

211: The peak number of malaria cases was based on which study or census?

216: Was there any statistical calculation to establish the ideal sample number? If yes, please describe clearly.

223: change the celsius degree sign.

251: Describe the age of minor participants more clearly. It's a little vague, just write smaller ones. The more detailed the better the work.

259: What is the average amount of blood collected in ul?

281: The RT-MqPCR part needs to be rewritten or more information needs to be added.

282: DNA or cDNA? Describe how the DNA was extracted. What kit is used? and What is the final extraction yield in uL?

282: What is the target of this PCR?

345: Clearly describe the age of children under one year in months.

353: Clearly describe the age of children under one year in months.

359: Clearly describe the age of children under one year in months.

378: add uL.

383: describe the age of children under one year in months.

386: I would like to see the RT-MqPCR results, with the sample amplification curves. To better analyze the results. I suggest adding these graphs in the supplementary results.

394: The frequencies in the text are not the same as those in table 3. Choose whether to use the number of samples or inhabitants.

457: Have epidemiological studies been carried out with anopheles in these regions?

1251: Improve the description of the legend in figure 1. Add what the green color on the map and the red balls mean.

1225: Define what is a title and what is a legend in figure 2.

1237: improve the description of the legend for figure 3. Describe what the color red means.

1261: Clearly describe the age of children under one year in months, in the group from < 1 to 10 years.

1277: It would be interesting to demonstrate the age and location of each positive sample.

1279: Rewrite the table in English.

Reviewer #2: Major Concerns

Overall, the manuscript presents us interesting data on the detection of asymptomatic cases of P. vivax on three distinct locations in a Comarca in Panama. And, according to the authors, this is the first ever report of asymptomatic cases in the Country. Therefore, the manuscript does hold scientific relevance.

However, apart from the Introduction, in the following sections, the text is increasingly repetitive and lacks a linear. objectivity. For example, the argument that asymptomatic infections have epidemiological consequences is repeated and rephrased in almost all sections of the manuscript. These repetitive arguments make the text unnecessarily long.

The data presented is based on the results of light microscopy and RT-MqPCR diagnostic tests from 551 asymptomatic people in three different localities. The take home message is that PCR over performs LM in the detection of asymptomatic cases - which was expected. The manuscript then tries to explore some population variables, such as sex and age, but the messages are not clear, specially since no statistical analysis is shown (not in the text and not even on the tables). Although the Methods section cites the Chi-square test, no clear indication of its use is presented.

The proper use of statistical analyses could help improve the manuscript as it would reveal whether any of the variables are indeed distinct among localities. If the authors have already performed the proper analyses, they should report them appropriately in the manuscript, specially in the tables.

The tables should be reformatted as well. Table 2 has terms to be translated into English. And Table 3 has too much information on display and without the results from the statistical analyses, some of these data are simply confusing, as for example, the many percentages.

There are several conclusions/inferences in the manuscript that are not backed by the data (specially without the statistical support) that the authors should consider removing. For example, this study is not suitable for inferences on sensitivity specificity of diagnostic tests, or for policy recommendations that children are more prone to asymptomatic infections.

In summary, my suggestion is that the authors revise the text making it more concise and (considerably) shorter, eliminating unnecessary repetitions and removing far fetched conclusions. Also, perform and report the statistical analysis on population variables.

Minor concerns

Pg11 L233 → positive LM OR PCR (not and)

Section Results, Analysis of the study population → The numbers and percentages are not adding correctly

Paragraph at line 351 → Why are CI values informed for a known exact number of people?

The title of table 2 says 3 communities, but the data is not presented by community

According to FIG4. Irgandi had the most cases in 2021, but had fewer asymptomatic cases. How is this interpreted?

Suggestion → reorganize TABLE 3 to have ML and RT-MqPCR as lines not columns

Discussion

"one of the most important variables that determines the intensity and prevalence of malaria in this region is the internal migration in the region of native people and the arrival of immigrants from South America who are symptomatic and asymptomatic carriers"

What is the migrant status of the participants? Are there more migrants in any of the communities?

Line 525 → These percentages are relative to which total numbers?

6. PLOS authors have the option to publish the peer review history of their article (what does this mean?). If published, this will include your full peer review and any attached files.

Reviewer #1: **Yes: **Dra. Gabriela Maíra Pereira de Assis

Reviewer #2: No

---

## [Author Response · Author response to Decision Letter 0]

5 Feb 2024

The corrections requested by the editor and reviewers were made.

---

## [Decision Letter · Decision Letter 1]

17 Mar 2024

PONE-D-23-20941R1Detection through the use of RT-MqPCR of asymptomatic reservoirs of malaria in samples of patients from the indigenous Comarca of Guna Yala, Panama: Essential method to achieve the elimination of malaria.PLOS ONE

Dear Dr. Carrera,After careful consideration, we have concluded that your manuscript has the potential to be published although some aspects of the manuscript will need to be changed prior to formal acceptance. Specifically, the language needs to be properly adjusted otherwise it might compromise the MS.

We look forward to receiving your revised manuscript.

Kind regards,

Luzia H Carvalho, Ph.D.

Academic Editor

PLOS ONE

Journal Requirements:

Reviewers' comments:

Reviewer's Responses to Questions

**Comments to the Author**

1. If the authors have adequately addressed your comments raised in a previous round of review and you feel that this manuscript is now acceptable for publication, you may indicate that here to bypass the “Comments to the Author” section, enter your conflict of interest statement in the “Confidential to Editor” section, and submit your "Accept" recommendation.

Reviewer #1: All comments have been addressed

2. Is the manuscript technically sound, and do the data support the conclusions?

Reviewer #1: Yes

3. Has the statistical analysis been performed appropriately and rigorously? 

Reviewer #1: Yes

4. Have the authors made all data underlying the findings in their manuscript fully available?

Reviewer #1: Yes

5. Is the manuscript presented in an intelligible fashion and written in standard English?

Reviewer #1: Yes

6. Review Comments to the Author

Reviewer #1: The manuscript is well written and includes all the requested corrections. The scientific findings are extremely relevant to the scientific community and health organizations.

Two additional corrections need to be made:

Table 1 - correct "Femen" to female

Table 2 - correct "porcentaje" to percentage

7. PLOS authors have the option to publish the peer review history of their article (what does this mean?). If published, this will include your full peer review and any attached files.

Reviewer #1: No

---

## [Author Response · Author response to Decision Letter 1]

30 May 2024

Doctor

Luzia H Carvalho

Academic Editor

PLOS ONE

Dear Doctor. Carvalho:

We hereby forward to you the corrections made to the manuscript [PONE-D-23-20941] - [EMID:91dfbf0d0b91cc2d] titled “Detection through the use of RT-MqPCR of asymptomatic reservoirs of malaria in samples of patients from the indigenous Comarca of Guna Yala, Panama: Essential method to achieve the elimination of malaria”, with corrections made according to the observations of the Plos One editor and evaluators. 

Editorial review:

1. The language was appropriately adjusted through Editage services, we attach a certificate of editing of the manuscript.

2. The list of references was reviewed and repeated ones replaced.

Reviewer 1

1. Table 1 - correct "Femen" to female

2. Table 2 - correct "porcentaje" to percentage

Thanking you for your attention, sincerely,

Dr. Lorenzo Cáceres Carrera

Senior health researcher at ICGES

---

## [Editor Report · Decision Letter 2]

3 Jun 2024

Detection through the use of RT-MqPCR of asymptomatic reservoirs of malaria in samples of patients from the indigenous Comarca of Guna Yala, Panama: Essential method to achieve the elimination of malaria.

PONE-D-23-20941R2

Dear Dr. Carrera,

We’re pleased to inform you that your manuscript has been judged scientifically suitable for publication and will be formally accepted for publication once it meets all outstanding technical requirements.

Kind regards,

Luzia H Carvalho, Ph.D.

Academic Editor

PLOS ONE
---

## [Editor Report · Acceptance letter]

18 Jun 2024

PONE-D-23-20941R2 

PLOS ONE

Dear Dr. Cáceres Carrera, 

I'm pleased to inform you that your manuscript has been deemed suitable for publication in PLOS ONE. Congratulations! Your manuscript is now being handed over to our production team.

Kind regards, 

on behalf of

Dr. Luzia H Carvalho 

Academic Editor

PLOS ONE